# Views of health professionals on risk-based breast cancer screening and its implementation in the Spanish National Health System: A qualitative discussion group study

Celmira Laza-Vásquez[1,2], Núria Codern-Bové[3,4,5], Àngels Cardona-Cardona[5], Maria José Hernández-Leal[6,7], Maria José Pérez-Lacasta[6,7], Misericòrdia Carles-Lavila[6,7], Montserrat Rué[8,9]*, on behalf of the DECIDO group¶

1 Department of Nursing and Physiotherapy, University of Lleida-IRBLleida, Lleida, Spain, 2 Health Care Research Group (GRECS), Lleida, Spain, 3 Escola Universitària d'Infermeria i Teràpia Ocupacional de Terrassa, Universitat Autònoma de Barcelona, Terrassa, Spain, 4 Health, Participation, Occupation and Care Research Group (GrEUIT), Terrassa, Spain, 5 ÀreaQ, Evaluation and Qualitative Research, Barcelona, Spain, 6 Department of Economics and Research Centre on Economics and Sustainability (ECO-SOS), Rovira i Virgili University (URV), Tarragona, Spain, 7 Research Group in Statistical and Economic Analysis in Health (GRAEES), Reus, Spain, 8 Department of Basic Medical Sciences, University of Lleida-IRBLleida, Lleida, Spain, 9 Research Group in Statistical and Economic Analysis in Health (GRAEES), Lleida, Spain

¶ Membership of the DECIDO group is listed in the Acknowledgments.
* montserrat.rue@udl.cat

**Data Availability Statement:** Data consists of recorded group discussions and the corresponding

## Abstract

### Background

With the aim of increasing benefits and decreasing harms, risk-based breast cancer screening has been proposed as an alternative to age-based screening. This study explores barriers and facilitators to implementing a risk-based breast cancer screening program from the perspective of health professionals, in the context of a National Health Service.

### Methods

Socio-constructivist qualitative research carried out in Catalonia (Spain), in the year 2019. Four discussion groups were conducted, with a total of 29 health professionals from primary care, breast cancer screening programs, hospital breast units, epidemiology units, and clinical specialties. A descriptive-interpretive thematic analysis was performed.

### Results

Identified barriers included resistance to reducing the number of screening exams for low-risk women; resistance to change for health professionals; difficulties in risk communication; lack of conclusive evidence of the benefits of risk-based screening; limited economic resources; and organizational transformation. Facilitators include benefits of risk-based strategies for high and low-risk women; women's active role in their health care; proximity of

transcriptions. Languages used were Catalan and Spanish. Data cannot be shared publicly because we did not ask participants' consent to share the recorded materials. Data are available from the The Drug Research Ethics Committee of the University Hospital Arnau de Vilanova at Lleida (contact via e-mail ceim.lleida.ics@gencat.cat) for researchers who meet the criteria for access to confidential data.

**Funding:** MR PI17/00834: Personalized breast cancer screening: assessment of its feasibility and acceptability in the National Health System. Instituto de Salud Carlos III and cofunded by Fondo Europeo de Desarrollo Regional (FEDER) 'Una manera de hacer Europa' https://www.isciii.es MCL PI18/00773: Collaboration of healthcare professionals to include shared decision-making in the breast cancer screening program. Instituto de Salud Carlos III and cofunded by Fondo Europeo de Desarrollo Regional (FEDER) 'Una manera de hacer Europa' https://www.isciii.es CLV Santander Program scholarship 2020: predoctoral fellow at the University of Lleida MJH-L: European Regional Development Fund (ERDF). European Union's Horizon 2020 research and innovation programme under the Marie Skłodowska-Curie grant agreement No. 713679 from the Universitat Rovira i Virgili (URV). The funders did not participate in the study design, data collection and analysis, decision to publish, or preparation of the manuscript.

**Competing interests:** The authors have declared that no competing interests exist.

women and primary care professionals; experience of health professionals in other screening programs; and greater efficiency of a risk-based screening program. Organizational and administrative changes in the health system, commitment by policy makers, training of health professionals, and educational interventions addressed to the general population will be required.

## Conclusions

Despite the expressed difficulties, participants supported the implementation of risk-based screening. They highlighted its benefits, especially for women at high risk of breast cancer and those under 50 years of age, and assumed a greater efficiency of the risk-based program compared to the aged-based one. Future studies should assess the efficiency and feasibility of risk-based breast cancer screening for its transfer to clinical practice.

## Introduction

Breast cancer is the most common cancer in women and overall. In 2020, more than 2.2 million new cases were diagnosed, representing 11.7% of all cancer cases. Breast cancer was the leading cause of cancer deaths in women with 68,000 deaths worldwide. In Spain, in 2020, the estimated numbers of incident cases and breast cancer deaths in women were 34,000 and 6,600, respectively [1].

Screening programs are one of the pillars in the fight against breast cancer, they aim to reduce mortality from this cause through early detection and treatment. However, there is evidence that screening may cause harm, e.g. false positive and false negative results or overdiagnosis of tumors that would otherwise not go on to cause symptoms or death [2], leading to negative consequences for women and economic consequences for health systems [3, 4].

Current population-based screening programs use age as the only condition to define the target population. With the aim of increasing benefits and decreasing harms, risk-based screening has been proposed as an alternative to age-based screening. Age, breast density, family history of breast cancer, previous benign lesions and genetic information have been the main risk factors used to define risk groups [5, 6]. Yanes et al. [7] showed that, in European populations, the addition of a PRS to the existing risk models improved their accuracy and predictive ability. Evans et al. [8] added a PRS and mammographic density to the Tyrer-Cuzick model, with the objective of enabling more targeted early detection/prevention strategies in population screening programs. The combined risk tool defined a low-risk group, about 30% of the total, such that cancers identified in this group were more likely to have a very good prognosis.

Although evidence of the effectiveness of risk-based screening from clinical trials is not available yet, a systematic review that included nine modeling studies and an observational study showed that with personalized screening, the gain in quality adjusted life years would be higher at a lower cost, compared with the standard age-based strategy [9]. However, evidence is lacking on feasibility and acceptance by the target population. French et al. [10] developed an automated system (BC-Predict) for offering an assessment of breast cancer risk and communication to women and health professionals. The study aimed to identify and resolve key uncertainties regarding the feasibility of integrating BC-Predict into the breast screening program in England. Pons-Rodriguez et al. recently published the protocol of a proof-of-concept study on feasibility and acceptability of personalized breast cancer screening in Catalonia (Spain) [11].

Although risk-based breast cancer screening seems promising, it is expected that its implementation as a population-wide screening will pose complex issues and will require the involvement of healthcare professionals. In addition, in a context of uncertainty where health professionals with clinical expertise have access to the best evidence and women have their experiences, values and preferences, it seems necessary to move from a paternalistic to a participatory style of care [12]. Shared decision-making would facilitate participatory decisions in clinical encounters where health professionals would explain, in a balanced way, the benefits and harms of screening [13] and provide risk-based screening recommendations.

Several studies have addressed the perspectives of health professionals on risk-based screening [14–19]. These works have described barriers such as the need for more evidence on effectiveness and efficiency [14], concerns about the capacity of the healthcare system to provide appropriate human resources, economic costs, lack of knowledge among healthcare providers [15], time constraints, low health literacy and language barriers to risk communication [18].

The organization of the Spanish National Health Service (SpNHS) is decentralized in 17 regions with screening guidelines set by the local governments and coordinated by the Network of Cancer Screening Programs and the Ministry of Health [20]. Most of the screening programs target women aged 50–69 years and perform biennial mammograms. However, opportunistic screening in women younger than 50 years is widely used, mostly in private consultations. Including women younger than 50 years in risk-based screening, so low-risk women are recommended to wait and high-risk women are screened, may improve the balance of benefits and harms, as mathematical models have shown [3].

The most recent recommendations of the Spanish Family and Community Medicine Society (SemFYC) state that the primary care physicians should provide objective and proven information, motivating women to make an informed decision about their participation in the screening program, and also should collaborate in the identification of women at high risk due to a family or personal history of breast cancer [21]. Therefore, studies exploring attitudes, experiences and opinions of healthcare professionals that may play a prominent role in implementing risk-based screening programs in our setting are needed.

The objective of the study was to explore the barriers and facilitators of implementing a risk-based breast cancer screening program from the point of view of health professionals, in the context of the SpNHS. The study is part of the DECIDO project, which aims to assess the acceptability and feasibility of offering personalized breast cancer screening and its integration into clinical practice [11].

## Methods

### Design and methodologic perspective

A socio-constructivist qualitative study on the barriers and facilitators of personalized breast cancer screening, in the Catalan Health Service, was conducted. Using a descriptive and interpretive study we intended to find out the meaning and interpretation of these barriers and facilitators, based on the meanings and explanations that health professionals attribute to them, according to their context, experience and reflection [22].

### Context and participants in the study

Field work was carried out in the cities of Lleida and Barcelona (Catalonia, Spain) between April and November 2019. The informants were health professionals (including health service providers, managers and administrative workers) from primary care, breast cancer screening programs, hospital breast units, epidemiology units, and clinical specialties such as oncology, radiation oncology, and diagnostic imaging.

Health professionals were selected according to a theoretical sampling strategy aimed at ensuring heterogeneity and capturing their potential discursive diversity [23]. The selection criteria were: work area in relation to breast cancer (early diagnosis, cancer treatment, and primary care), field of expertise (nurse, midwife, general practitioner, specialist or health manager), and geographical health area (Lleida, Bellvitge Hospital/Catalan Institute of Oncology, Parc de Salut Mar, and Vall d'Hebron Hospital).

The participant search was carried out by the research team. We contacted the heads of the health care services to inform them about the study and ask them to inform and request contact consent from health professionals working with them. Professionals interested in participating received information about the study, dates of the discussion groups, and were invited to send a reply on their decision to participate. The sample included 29 health professionals whose characteristics are described in Table 1.

## Data collection

Discussion groups were held to collect the participants' perspectives. A discussion group consists of a meeting of seven to ten people who have been selected based on specific profiles

**Table 1. Characteristics of the study participants.**

| Discussion group (DG) | Participants (P) | Work area | Health service area | Professional profiles |
|---|---|---|---|---|
| DG1 | P1 | Primary Care | Lleida | Nurse |
| | P2 | Hospital Breast Unit | | Nurse |
| | P3 | Radiation Oncology | | Oncologist |
| | P4 | Primary Care | | Midwife |
| | P5 | Primary Care | | Physician |
| | P6 | Catalan Health Service | | Management |
| | P7 | Hospital Breast Unit | | Surgeon |
| | P8 | Hospital Breast Unit | | Psychologist |
| DG2 | P1 | Hospital Breast Unit | Lleida | Management |
| | P2 | Primary Care | | Nurse |
| | P3 | Breast Cancer Screening Program | | Management |
| | P4 | Radiation Oncology | | Nurse |
| | P5 | Primary Care | | Physician |
| | P6 | Hospital Breast Unit | | Nurse |
| | P7 | Radiation Oncology | | Nurse |
| DG3 | P1 | Hospital Breast Unit | Barcelona | Oncologist |
| | P2 | Diagnostic Imaging Service | | Radiologist |
| | P3 | Primary Care | | Physician |
| | P4 | Breast Cancer Screening Program | | Management |
| | P5 | Hospital Breast Unit | | Nurse |
| | P6 | Epidemiology and Evaluation Service | | Physician |
| | P7 | Primary Care | | Management |
| DG4 | P1 | Early Detection Unit | Barcelona | Radiologist |
| | P2 | Catalan Health Service | | Management |
| | P3 | Primary Care | | Physician |
| | P4 | Early Detection Unit | | Psychologist |
| | P5 | Breast Cancer Screening Program | | Physician |
| | P6 | Breast Cancer Screening Program | | Nurse |
| | P7 | Primary Care | | Management |

according to the research objectives, to comment and debate on a series of topics, induced by a moderator. It is intended to reproduce a micro-social situation in the image of what would be a macro-social situation, through the interaction of the different profiles, as an effective way of extracting experiences, reactions, emotions and ideas according to Patton et al. [24].

The intention was to build a debate based on an initial provocation (e.g. is it possible to implement a risk-based cancer screening program in the SpNHS?). The discussion group technique is less directive than the focus group, where the moderator has a guide of questions for organizing a discussion that are answered in a timely manner [25]. We intended to build a joint and open conversation from the discursive confrontation between the participants [23]. The discussion group provided a presentation and guided discussion about barriers and facilitators of implementing a risk-based breast cancer screening program from the perspective of health professionals. The list of open-ended questions, presented in S1 Table, was developed from a literature review. Each discussion group had a moderator with extensive experience in qualitative research (ACC or NCB) and an observer from the research team (MR or MC).

Four discussion groups were held, two in Lleida and two in Barcelona. Discussion groups were held at the Medical School of the University of Lleida and the headquarters of the Catalan Institute of Health in Barcelona. The team previously discussed and agreed on how to conduct the groups. Before starting the discussion group, the objectives of the study were presented and the concepts of personalized screening and shared decision-making were introduced. A brief summary of a proof of concept study about the feasibility and acceptability of risk-based screening, carried out by the DECIDO team, was presented [11]. Informed consent and permission for audio recording of the session were obtained. Then, the discussion was started by the open questions through the guided discussion (S1 Table). The duration of the discussion was approximately 90 minutes. At the end of the group discussion ACC and NCB took notes. The sessions were audiotaped and transcribed verbatim by two transcriptionists, not members of the study team, and later reviewed by ACC and NCB. The research team decided to end the data collection when the information obtained was considered sufficient to answer the research questions [26]. Transcripts of discussion groups were not returned to the participants for comments.

## Data analysis and interpretation

The textual corpus made up of the discussion group transcripts was analyzed by two researchers who performed a descriptive-interpretive thematic analysis [27]. The entire research team participated in a session to interpret the results.

The transcript analysis consisted of a) reading and rereading the transcripts; b) reviewing the field work notes and writing the first intuitions and pre-analytic thoughts; c) coding texts and extracting categories in a mixed way (inductive and deductive) considering the objectives of the study and Légaré's taxonomy that categorizes barriers and facilitators as knowledge, attitudes and behaviors [28]; d) reconstructing the categories by performing a work of contrast with the texts and between the texts, concept maps and analysis of co-occurrences between the discursive profiles and the barriers and facilitators identified in the analysis, in order to achieve an explanatory framework of the meanings according to the different discourses. The analysis was carried out with the support of the Atlas-ti version 8.4 software [29].

## Quality and rigor

To ensure the validity of the study, different strategies were taken into account: discussion of the suitability of the strategy to generate information, theoretical sampling to ensure discursive variability, conducting four discussion groups to reach information saturation, and triangulation in the analysis and discussion of the results by the entire research team [30].

The participation of the entire research team fostered reflexivity throughout the development of the study, by discussing prior knowledge and experiences and explicitly describing preconceptions that were considered in both the design and interpretation of the results [23].

### Ethical aspects

The Drug Research Ethics Committee of the University Hospital Arnau de Vilanova at Lleida approved the study. All the informants, after having been informed of the objective of the study and the involved institutions, participated voluntarily. The sessions were recorded guaranteeing the anonymity and confidentiality of the data. Thus, the identities of the participants and the references that could identify them were anonymized.

## Results

Fig 1 summarizes the visions of healthcare professionals on the implementation of risk-based screening for breast cancer. Barriers and facilitators have been grouped in four categories, related to 1) women; 2) health professionals; 3) the risk-based screening program; and 4) the health system. Some organizational proposals about how to implement a risk-based program are listed at the bottom of Fig 1.

### Barriers and facilitators, related to women, for participating in personalized risk-based breast cancer screening

According to the informants perspective, the main barrier for women to participating in risk-based screening would be resistance to having fewer mammograms, for women at low-risk (Table 2). More than 30 years of promotional efforts to encourage participation in screening programs and intense media coverage of breast cancer have raised awareness of the disease. This can make it difficult to understand that a reduction in the number of mammograms can be beneficial, which may cause some women to reject the personalized screening

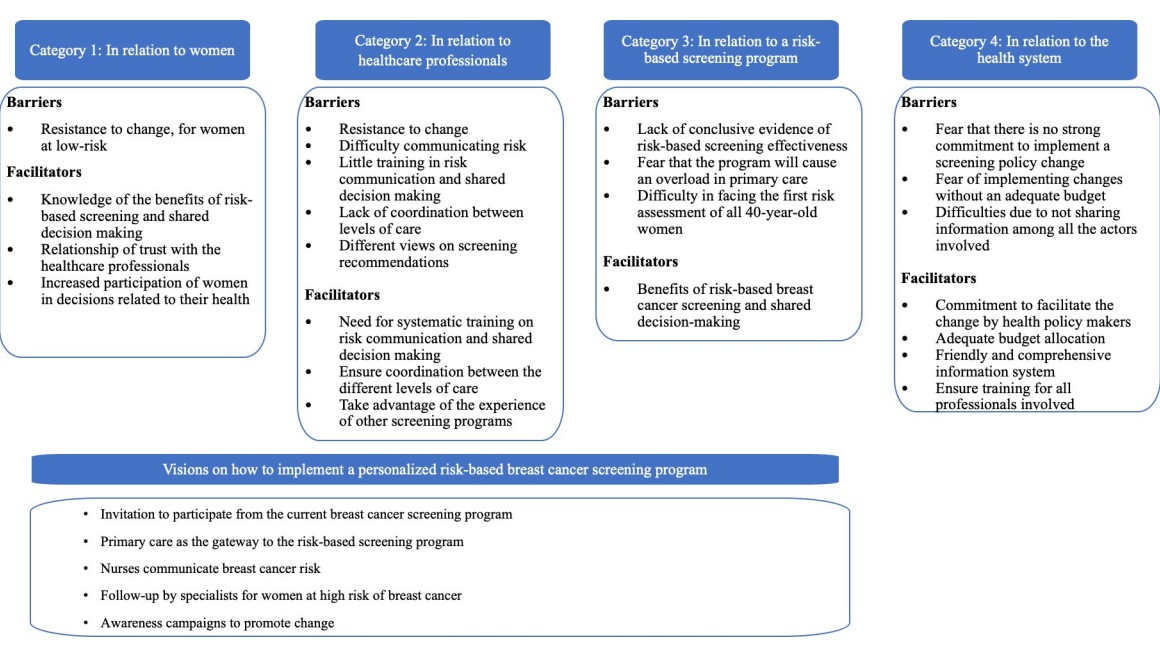

**Fig 1. Visions of healthcare professionals on the implementation of risk-based screening for breast cancer.**

**Table 2. Selected quotes from health professionals' discussion groups on barriers and facilitators for women regarding participation in risk-based screening.**

| |
|---|
| **Barriers: Resistance to change** |
| The barriers are that a woman believes that a mammogram should be done every two years, or her private gynecologist has recommended she have a mammogram, and what she does is either go to the private gynecologist or the screening program. This woman, no matter how much you tell her it should be every three years because her risk is low [. . .] (DG2P3) |
| **Barriers: Rejecting the personalized screening recommendations** |
| But, on the other hand, those who are used to biennial screening from the age of 50 onwards are now being told "well no, since your risk is low, in your case it will be every three years". How will this woman accept this? (DG1P4) |
| A barrier: to be told "you have to have a mammogram every three years" and then the GP tells you "if you who have private insurance, do it at the clinic" (DG2P5) |
| **Facilitator: benefits of a risk-based program for young women** |
| [. . .] But there is increasing demand from young women. They don't know if they are at risk or not, if there is a family history then obviously look, look to the private system or primary care or whatever. (DG1P2) |
| I think it is good that a part of the population that now is not covered, ages 40 to 49, will have one mammogram every three or two or one years. I think that's a facilitator (DG4P5) |
| **Facilitator: proximity and trusting relationships with women** |
| [. . .] Fortunately women have a good relationship with nurses and doctors, especially primary care or specialists and women kind of do what they tell them to do, right? [. . .] (DG1P6) |
| Yes, yes, that would be a facilitator, every primary care doctor has their patients and can inform them about this program. (DG2P1) |
| **Facilitator: growing interest and proactivity of women in their health care** |
| And I think that women, who are already very proactive towards their health, if this decision can be made in an informed way, deciding for or deciding against, we will make the whole system more efficient. (DG3P1) |
| [. . .] And women have become empowered in breast cancer screening, in prevention because we know that if we detect it, we have better survival. And there are women in their 40s who come to ask us . . . This is why screening is so successful too. (DG1P3) |
| **Facilitator: intensive media attention to breast cancer** |
| Breast cancer now has become visible, on a social level, on a communication level, there is a lot of talk, a lot of famous people have said they have breast cancer, breast cancer is no longer a taboo [. . .](DG1P3) |

recommendations and opt for using the private health system to ensure having annual or biennial mammographic exams.

Facilitators that could reduce the impact of this barrier focus on the benefits of a risk-based program compared to "one size fits all." A risk-based program would entail a) an initial risk assessment and a potential start of screening at a younger age, something that is not possible now, within the public health system, but many women would like, and b) more frequent exams and preventive measures for high-risk women. The participants expressed the need to inform women adequately in order to increase acceptance and participation. They also considered a facilitator the proximity and the relationship of trust that women have with their primary care professionals (PCP). An active role for PCPs could involve informing and inviting women to participate in the program. Participants also mentioned the growing interest and proactivity of women in decision-making related to their health, which may be a consequence of the strong diffusion of breast cancer burden through mass media. This has raised awareness among women and has increased their interest in early detection of the disease.

## Barriers and facilitators associated with health professionals, for being involved in personalized risk-based breast cancer screening

With regard to barriers, the informants mentioned that even though there is evidence in favor of risk-based screening, there would be resistance to shifting from a model that has been in use

for more than 30 years to a new one that requires significant changes in current practice and policy (Table 3).

Professionals were aware of the difficulty involved in communicating risk to women or developing shared decision-making so that women could decide whether or not to participate

**Table 3. Selected quotes from health professionals' discussion groups on barriers and facilitators for health professionals related to participating in risk-based screening.**

**Barrier: resistance to change**

Of course, for example, from primary care what I see are the barriers set up by the professionals themselves. As a woman, I am more aware, but my colleague who is 50 years old perhaps is more aware of other aspects and not so much that. The barriers of the professionals themselves, one's own resistances. That's hard to work with. (DG2P5)

[. . .] cultural change within primary care physicians will be complicated. I don't know, I mean, I'm optimistic from the study point of view, of people participating eagerly. But I don't see this as an easy change! [. . .] There are family doctors who do not want to get into this game because they do not want more work (DG3P7)

**Barrier: difficulties in risk communication**

Now please change this message, this conversation to Urdu. Thirty five percent of the migrant population in our center [speak Urdu]. It is very complicated. (DG4P7)

The issue is how society perceives these risks and how they are communicated! And obviously, it is true that here it is necessary to incorporate somebody with the ability to communicate risk, right? And knowing how to communicate positively and negatively, using the right words [. . .] (GD3P1)

**Barrier: professionals do not have training in risk communication and shared decision-making**

[. . .] the issue of shared decisions, I think it's an issue that will prevail, and therefore not just with this, but with everything! However, we the professionals are not sufficiently trained, maybe because [. . .] or we do not have instruments or we do not have time or we still have to develop it more. (DG1P6)

**Barrier: differing views on breast cancer screening recommendations**

[. . .] we all know that screening has to be done every two years and that is what all doctors have to recommend to the patient. And there are doctors who say, "I would do it every year." In other words, it is not clear that all professionals agree, and breast cancer screening has been going on for more than 20 years! (DG1P1)

We may find that we tell a woman that she has a risk level that requires a mammogram every 3 years and another professional comes and says: "no, no, I would do it every year". We are going to find that, for sure! (DG1P3)

[. . .] we run into a wall, if I go to the gynecologist and as they tell me "since you have a mother who had a breast neoplasm, you should have had at least one mammogram before 50, at least one". I was 47 and I said yes. But that's not it, is it? Messages can't be like that. (DG4P7)

**Facilitators: training of health professionals**

Because, of course, they need specific training. Either give them a series of very clear items, and say you have to look at this, this and this [. . .] (DG1P7)

**Facilitators: levels of care coordination**

[. . .] I believe that a risk-based program could never be done without the participation of all the people involved and a good information system behind it and good planning and estimation of needs, techniques, professionals [. . .] (DG3P4)

Unify criteria. Please unify criteria, because the clinical guides say one thing, European urology says another, Catalan urology says another and Americans are saying another. Then, I am very sorry but I think it is one of the hardest battles. (DG4 P7)

**Facilitators: receiving feedback and information on the program outcomes**

Maybe it would also be good to see the result of all the work they have done, makes it more motivating to the professional, "well I have dedicated many hours to it, but I see the result". This is important. (DG2P2)

And then there's something super important, which has worked very well for us in colon screening, aside from training, it's the feedback of the work done. If you give feedback to people, of the 400 women you have, in your clinic, who are 40 to 70 years old, these have done genomics and those have been diagnosed, this is not a lot of work for the professional, it doesn't cost him anything, what costs is "do this" and you don't give any feedback. (DG2P3)

**Facilitators: lessons and experiences from other screening programs**

In the colorectal screening we have worked very well because of the proximity of the patient to the healthcare professional [. . .] And people say, "I have a health problem, I go to Dr. X, who is my doctor". (DG2P3)

[. . .] The "Cervix" went from [recommending] annual cytology to recommending every three years and there was quite a significant resistance from the population. It is a matter of informing well, explaining well [. . .] (DG3P4)

in screening. Thus, professionals of diverse profiles stated that, despite the fact that the estimated breast cancer risk can be considered an objective measurement, there are multiple aspects that a woman will consider, such as individual experiences and their environment. Participants called attention to the fact that most professionals do not have training in these subjects, turning this circumstance into a barrier.

Another barrier raised by the informants is the differing views on breast cancer screening recommendations among health professionals, which could also be the case if personalized screening was implemented. That would result in women not being offered the established recommendations and could cause confusion.

Participants raised some facilitators to mitigate the impact of the previously identified barriers. Among them, provision of systematic training over time to professionals involved in the program, especially those working in primary care. They also highlighted the importance of coordination of the different levels of care (primary care, current screening program, hospital services) and receiving feedback and information on the program outcomes. An information system shared by all the actors and an adequate allocation of human and economic resources were also considered important facilitators. Finally, the informants suggested building on the learnings and experiences of other existing screening programs, such as cervical or colon cancer, in which PCP or even pharmacists have participated.

## Barriers and facilitators related to implementing a risk-based screening program

The informants considered the uncertainty related to the lack of conclusive evidence on the effectiveness of risk-based screening and insufficient accuracy of risk measures to be an important barrier (Table 4). Some participants emphasized that more research is needed before implementing risk-based screening. This barrier may contribute to resistance to change, and possibly both barriers feedback mutually to one another. A second barrier, signaled mostly by PCP, arose in relation to the hypothetical scenario that the program burden fell mainly on them. The informants, in general, agreed that primary care should be the gateway to the screening program and also the setting where risk is communicated and recommendations are made. However, PCP expressed concern and some even rejection, due to lack of time and job instability, unless relevant organizational changes were made. Finally, the participants noted the high costs of measuring risks for a large number of women in the initial phase of program implementation.

On facilitators, informants stated the above mentioned benefits plus increasing accuracy in risk quantification, implementing shared decision-making, and allocating more screening resources to the high-risk group for a higher efficiency. These seemed like strong arguments for informants to justify the need to move toward risk-based screening.

## Health systems barriers and facilitators, for implementing personalized risk-based breast cancer screening

Since the potential implementation of risk-based breast cancer screening will require extensive changes in current practice, the informants expressed that the change would not be possible without the commitment of health policy makers and an adequate budget allocation for program implementation and maintenance (Table 5). In fact, the extra amount of financial and human resources for the implementation of the program is an economic barrier. Another barrier mentioned was the set of limitations of the current information system regarding the real-

**Table 4. Selected quotes from health professionals' discussion groups on barriers and facilitators related to implementing a risk-based screening program.**

| |
| --- |
| **Barrier: lack of conclusive evidence on the effectiveness of risk-based screening** |
| [. . .] In order to introduce screening at the population level, a series of randomized clinical trials were carried out, it had to be shown over and over again that mammography is effective. To introduce personalized screening we are also going to need to provide convincing scientific evidence. And currently, what is the strong evidence to make a strong recommendation in favor of personalized screening over standard screening? Or instead of opportunistic screening? Or instead of not screening? At the moment I think, as far as I know, that the studies are underway, but there are no results. So, at this moment, it would be a barrier for me if I wanted to convince the people that I have around and they tell me: "hey, I already have a lot to do, why are you asking me to change?" (DG3P6) |
| **Barrier: program burden may fall on primary care** |
| One thing is the colon screening where you collect the sample and the other thing is the genomic test. The genomic test cannot be performed at home by the user, so they need assistance. Overloading primary care with more, there are 50,000 women and I don't know what percentage will decide to not participate, but if we have 20% who will say no, we will have 40,000. (DG2P3) |
| [. . .] is a burden for primary care, probably unbearable from the current situation . . . If you are the primary care physician and you have to invite women, you need time or infrastructure . . . I understand [. . .] (DG4P5) |
| **Barrier: lack of time and job instability of primary care professionals** |
| And, I'm talking from primary care . . . There's a lot of nursing rotation . . ., monthly contracts and such, and now I'm going into policy a little bit. So what then? [. . .] I cover several consultations . . . And our nurses are changing practically every month. (GD1P1) |
| [. . .] First, all these doctors have to be trained, and that's not easy. I say this because I've tried it several times with breast topics and I haven't quite gotten it, [. . .]. And then there's medical staff rotations, meaning you have the trained medical staff and over the holidays there are staff changes . . . (DGP2P1) |
| And it is also the lack of time, in primary care, what I see as the biggest problem, personally of course, is time. (DG2P2) |
| **Barrier: high costs of measuring risks** |
| The problem is that all, all women from the age of 40 onwards have to have a mammogram, the clinical history, the genomic study, plus the visit of the doctor or nurse who has to explain all these risks to you, economically I do not know if the health system can currently assume this for all women of 40 years of age. (DG1P3) |
| [. . .] There would also be a resource that has not been considered which is the laboratory. There will be hundreds of SNPs to evaluate. (DG3P4) |
| **Facilitator: increased accuracy in risk quantification** |
| . . . the more the screening program is adapted to the probability of having cancer, the better it works. Therefore, you would remove women with very low probability and improve the precision for women with very high probability. I think conceptually, yes. (DG3P4) |
| What happens is that perhaps this more accurate estimate of risk is not all of a sudden. I think doing this would be a huge step in the accuracy of risk, which is not being done anywhere in the world. (DG4P4) |
| **Facilitator: higher efficiency of risk-based screening** |
| I think screening with mammography has reached its ceiling [. . .] And there is a need for change [. . .] it will no longer be considered beneficial because [. . .] it is already being demonstrated [. . .]. In addition, we have tools that allow us to better estimate risk. [. . .] And you give value to the test and also to the risk estimate. This is very important for adherence! (DG3P1) |
| [. . .] 30% of breast cancers we treat in Lleida are in women less than 50 years old and 30% are over 70. Therefore, we have 60% of the population outside the screening program. The impression we have is that there are patients, people, women, who do not need to do a biennial screening program and instead, there are women who are outside the age range who will surely need it. And I think it's a very, very interesting program. (DG2P1) |

time integration of screening data and clinical records, as well as the access of all health professionals involved in the screening program.

Considering the identified barriers, the facilitators that would reduce their impact are, on the one hand, the commitment of health policy makers to implement the program with its corresponding budget and human resources allocation, and on the other hand, a friendly and comprehensive information system that connects the different actors involved and facilitates communication between them.

**Table 5. Selected quotes from health professionals' discussion groups on health system barriers and facilitators.**

| |
|---|
| **Barriers: commitment of health policy makers** |
| [. . .] the problem comes from bureaucracies and administrations (DG2P3) |
| [. . .] another barrier for me, and more in what we are discussing, is the decision of the Department of Health [. . .] (DG3P7) |
| **Barriers: adequate budget allocation** |
| Well, I think that's the first requirement, right? You cannot initiate a policy without a budget. It is important. (DG3P2) |
| [. . .] The problem I see most as the main barrier is: who pays for these mammograms? It's basically . . . Who pays for everything? (DG4P3) |
| **Barrier: limitations of the current information system** |
| Yes, I think the data system we have is a barrier. There are people who don't get the invitation . . . People who come to you for a consultation and say "I have not received anything about the mammography" or "I have not" and you try to redirect them to the programs, but we see this with a certain . . . frequency, because of database problems. (DG4P7) |
| [. . .] the whole subject of information technology in our environment is very slow, but not impossible. Different hospitals have different information systems. I think it's good to be determined and optimistic but it's complicated (DG3P7) |
| **Facilitators: commitment of health policy makers** |
| If there is money and there is a good information system, I think the population benefit will be impressive. (DG2P3) |
| **Facilitators: friendly and comprehensive information system** |
| There is a basic issue [. . .] a very well-designed information system would be needed. With a good computer program that makes a classification of the risk groups, I think the assessment is not complicated and can be done. (DG2P1) |
| [. . .] in the same way that you have a tab that says "clinical course", "documentation" or "diagnostic tests", I think that for any woman over 40 years of age a new tab should appear with " breast cancer risk" [. . .] And, there, the information could be updated, each time there is a change. (DG3P1) |

## Organizational proposals for the implementation of a personalized breast cancer screening program

The open questions were: *How should the personalized screening program be organized*? *Who would do the different functions*? There was no consensus on how to organize a personalized screening program and there was controversy on the role of professionals that work at the current screening programs and in primary care. However, the section has limitations as it was not deepened enough due to lack of time. The moderators were strict on the planned time in order to prevent the informants fatigue.

First, the informants discussed who should and how to make the first contact for informing and inviting women to the program (Table 6). They proposed that women be invited by primary care physicians and nurses in clinical encounters or by a breast cancer screening program by mail or phone. They also suggested that more time and adapted decision aids would be necessary for women with a low educational level and/or language barriers.

Second, they debated risk measurement and risk-based screening recommendations. The informants emphasized the importance of communicating individual risk in plain language, to improve understanding and decision-making. They proposed that primary care physicians and nurses communicate risk and make screening recommendations, given that most women have a trusting relationship with them. High risk should be communicated by physicians whereas moderate or low risk could be communicated by nurses. In fact, there was a general consensus on the role that nurses could play in personalized risk-based screening.

Third, the discussion dealt with *who and how to perform the individual follow-up*. Informants considered that, with a good IT system, it could be done either by the screening program or

**Table 6. Selected quotes from health professionals' discussion groups on organizational proposals for a risk-based screening program.**

| |
|---|
| **Invitation to women** |
| [. . .] it is important that we think that in primary care there are family doctors and nurses. [. . .] So, I think that at the women's first visit, women's questions, mammogram referral, taking saliva samples, etc., initial explanation of the program and concepts of overdiagnosis and quaternary prevention, I think a nurse could do that perfectly. (DG4P3) |
| First, everyone has to be involved, all the actors, primary, specialized, and radiology also thinking in the same way, and then develop a software that maintains it . . . which would pose two things: one, how do you approach the population. That is, how do you call them, if you wait for them to come or you call them, and once they have come there is also something that has not been considered; the family doctor sees it, but someone has to organize the agendas, considering that you have this entire population, where one needs an annual mammogram, another every three years, another every two [. . .] In other words, technical offices will be necessary, but we can be an actor who organizes it so that it is primary care who makes the contact, which would be an ideal solution, perfectly trained nurses could do that, and people who simply organize. And don't worry, I would already be behind for [. . .] (DG3P4) |
| **Risk measurement and risk-based recommendations** |
| I imagine it as inviting a patient to a first visit . . . near radiology, where someone does the saliva extraction to assess the SNPs, the first mammogram, the radiologists classify the density in the same way, so that the measurement is also accurate, that everything goes into a database where SNPs, the density and the demographic questionnaire are added, and the risk assessment of a first contact with a professional, be it a nurse or a family doctor, that explains it to her. (DG3P4) |
| [. . .] and once the risk results arrive, with a leaflet or decision aid, if it's negative and very low risk, a nurse could do it. If . . . a little more risk, maybe then it's worth it for the doctor to explain it, right? (DG4P3) |
| **Who and how to do the follow-up?** |
| [. . .] once there is an estimated risk it is obvious. I mean, if a woman has to do it every year, she has to do it every year. And the one every three years, too. Then the thing would be that the technical office, at this time, the screening program would act to schedule women and mammograms, because the reading task would already be done by the trained radiologists. (DG2P3) |
| I understand that one thing is the admission into the program and identifying women and stratifying them by risk, I understand that this is a starting point, but then all the invitations of the following successive exams every year, every two years, every three or every five, I understand that another entity does it or it is the same primary care that is already inside the wheel. (DG4P3) |
| **Need of educational programs targeted to health professionals and women** |
| Let's make an ad on TV and in many newspaper articles. People are already beginning to understand. (DG4P3) |
| Informative materials should be adapted, there should be materials translated into other languages and culturally and educationally adapted, the explanation for people with a lower level of education. (DG2P2) |

PCP. If primary care managed the follow-up, additional resources for this health level would be needed. For high-risk women, breast specialists should inform and follow them up.

Finally, the participants indicated the need for educational programs targeted to health professionals and women, including mass media campaigns for large audiences. They expressed their confidence in these interventions to reduce resistance to change and facilitate the acceptance and participation of women.

## Discussion

This study explored, using a qualitative approach, the barriers and facilitators to the implementation of a risk-based breast cancer screening program as well as some organizational proposals for its implementation, from the point of view of a diverse group of healthcare professionals working in the SpNHS.

### Barriers and facilitators related to women

The participants expressed concern about a reduction in the number of screening mammograms for low-risk women, a finding consistent with other studies. In the UK, Meisel et al. [31]

explored attitudes towards modifying frequency of mammography screening based on genetic risk. They found that although 65.5% of the women supported the idea of varying the screening frequency, only 58.8% were willing to reduce mammographic exams if they were found at lower risk. Instead, 85.4% were willing to increase the frequency if at higher risk. A study of European women's perceptions found that more intensive screening for women with above average risk was generally welcomed, but screening recommendations for the other risk categories were met with skepticism [32]. Another study, which assigned participants to one of four hypothetical breast cancer risk scenarios (low, average, moderate, or high), with subsequent screening and prevention recommendations, found again that women's level of acceptance depended on their assigned risk category [33]. Only 13.1% of the women assigned to the low risk scenario found a 4-year screening interval acceptable, with 27.4% of these women opting for supplemental mammography screening outside of the national screening program.

Several authors have discussed the reasons for this barrier. In a qualitative study with North American women [34], personal acceptance of risk-based screening was mixed. While some believed that risk-based screening could reduce harms, others thought screening less often might result in missing a dangerous diagnosis, and many expressed concerns about the feasibility of risk-based screening and questioned whether breast cancer risk estimates could be accurate—either because women did not remember risk factors or were confused about how risk would be calculated. Some women also suspected that risk-based screening was motivated by a desire to save money rather than reduce harms. Dutch women described the role of perceived risk in the acceptability of personalized exams. They believe that if their estimated risk does not correspond to their perceived risk, they will be less likely to accept the screening advice [32].

In the UK, McWilliams et al. [35] explored healthcare policy decision-making stakeholders' views on a low-risk breast screening pathway. Participants identified individual beliefs about risk and knowledge of breast cancer and screening as key factors that impact women's responses to low-risk stratification. Among uncertainties that need to be resolved before implementation, they signaled accurate identification of low-risk women, gaining women's acceptance, and having evidence of lack of harm. In a study with professionals, Rainey et al. [36] suggested that women will question their personal risk information and their assigned pathway of care (screening frequency or biomedical prevention) and they expected an increase in opportunistic screening, particularly from women at low risk. As some authors suggest, there is a need for a) educating women on the benefits and harms of risk-based screening and prevention, to facilitate acceptability and informed decision-making [16], and b) developing effective communication materials to minimize resistance to screening reduction for those at lower risk [31]. A research synthesis on factors associated with the acceptability of human papillomavirus HPV testing for cervical cancer found that strategies that increase women's knowledge and efforts to increase health care providers' awareness might also increase acceptability of wider screening intervals [37].

Unlike other studies [16, 38, 39], participants in our study did not consider the potential discrimination that women may suffer from the use of their individual risk data for employment issues or health insurance discrimination to be a barrier. The universal health coverage of the SpNHS with free access to health care may explain this result.

Our study participants identified several facilitators, related to women, for participating in a risk-based screening program. First, and consistent with Puzhko et al. [14], they considered that a risk-based screening program is beneficial for women, especially for those at a higher-than-population average level of risk and for those who have objective (e.g. family history) or subjective reasons to be concerned. Thus, for some women, knowing the risk would be reassuring and would reduce anxiety, which would facilitate their participation in the program and in shared decision-making. This, in turn, could reduce the inappropriate use of

opportunistic screening [14] and increase the attendance at subsequent screening exams [40]. However, Puzhko et al. emphasized that women may not truly benefit from risk-based screening unless their health professionals understand the risks and benefits of screening and can interpret the results adequately [14]. Second, several studies suggest that a healthcare professional (i.e. primary care physician or nurse) should be involved in the invitation, risk stratification, risk communication and follow-up of women in a risk-based screening program [41]. As participants in our study mentioned, a relationship of trust with the health professional [14], as well as relying on the experience of risk communication acquired from other health conditions in preventive medicine [42], would facilitate a personalized approach [43]. Third, the increasing participation of women in decisions that affect their health represents an incentive to participate in a personalized screening program. Our results agree with those of other studies reporting risk knowledge as an opportunity to have a proactive attitude [32], by being aware of the impact that breast cancer could have on women's lives [44]. Likewise, health professionals value the proactive approach to shared decision-making, as women can take control of some of their risk factors, take steps to reduce them and try to avoid breast cancer, all findings that are in line with women finding risk based screening acceptable [36].

## Barriers and facilitators associated with health professionals

Our results showed greater knowledge and a more positive attitude on the part of professionals towards personalized screening when compared to a previous study conducted in 2016 [45]. In line with this view, Puzhko et al. [14] reported that health professionals acknowledged the substantial benefits of the risk-based program. However, our discussion groups, as others have reported, expressed concern about the critical view of health professionals as a result of the uncertainty in the evidence on the safety and cost-effectiveness of the risk-based approach [35], and the resistance to change of professionals, reluctant to modify their established routine.

According to our study, training is essential for successfully implementing a program. Our results are consistent with other studies that identified professionals' concerns about lack of competence to communicate risk and subsequent screening recommendations effectively [14–16]. These authors suggested implementing educational programs that address these knowledge deficits, for each woman and risk category. Our study participants discussed how to implement training in the primary care setting, taking into account the diversity of settings (urban versus rural) and the mobility of health professionals. They also proposed strategies addressed at improving their training.

The increase in complexity of risk-based screening will require collaboration at multiple levels of care. Although Rainey et al. [16] suggested that the countries that already have a population-based screening program may experience fewer difficulties implementing risk-based screening, in our country it will be necessary to improve collaboration between primary care, population-based screening programs, and also specialized care for high-risk women.

Moreover, all discussion groups considered it essential that Health Authorities incorporate the personalized screening program into a health policy goal with adequate allocation of human and economic resources. Only then, dissemination of the personalized program, women's risk assessment, risk communication and levels of care coordination can be guaranteed. Without a health policy commitment and budget allocation, professionals perceive that there will be resistance to change. These results are concordant with previous studies. Whereas Esquivel-Sada et al. [15] pointed out the lack of confidence of professionals in the public health system to provide an adequate allocation of human and economic resources, McWilliams et al. [35] stated that risk stratification is potentially profitable and consequently, further study of the cost-effectiveness around personalized risk-based programs will be necessary.

## Barriers and facilitators related to implementing a risk-based screening program

As a barrier, participants mentioned the uncertainty around the effectiveness of risk-based screening. Their doubts relied on the lack of evidence of benefits from a risk-based model compared to the current age-based program. To advance in this area, the European Collaborative on Personalized Early Detection and Prevention of Breast Cancer (ENVISION) network [46] brings together several international research consortia, from 19 countries, working on different aspects of personalized early detection and prevention of breast cancer. The network highlights how risk stratification has improved with the use of comprehensive models incorporating genetic and epidemiological risk factors and mammographic breast density, which have shown an excellent calibration for the European, Hispanic and African-American population [7]. Recent studies report that PRS models derived from women with European ancestry for breast cancer risk generalized well for women with European and Latina ancestries and to a lesser degree to women with African ancestry where further studies with larger sample size are needed [47, 48]. Although research is still ongoing to assess the clinical utility of PRS for population screening programs, Yanes et al. point out that polygenic testing is already being implemented in specialist familial cancer clinics to provide additional information for women with family history and uninformative genetic test results [7].

On the one hand, despite the doubts expressed by professionals as a result of the lack of evidence of the effectiveness of risk-based screening, they recognized that it would provide greater accuracy in measuring individual risk. Thus, health professionals seemed to have a good predisposition towards risk-based screening. However, as Petrova et al. [17] propose it is essential that they know the results of the research and understand the statistics of early detection in order to avoid misleading and potentially harmful recommendations. Additionally, the general population should be exposed to media campaigns that inform them about the pros and cons of a risk-based program [42].

On the other hand, the barriers that most concern our professionals come from the organizational system, in particular the fear of an increase in the workload for PCP and the lack of economic resources. These impressions are based on the fact that the implementation of the personalized program involves a more complex organization and more agents involved [16] such as patients, providers, facilities, health care systems, and regional and national policy-making organizations [43, 49]. In addition, our study participants assume that there is an increase in costs derived from the genetic information in risk measurement and the time spent by professionals in communicating risk and shared decision-making. These costs would be largely offset by the increased effectiveness and efficiency of risk-based screening [3, 9, 50–53], as well as by greater well-being provided by fluid communication between women and professionals regarding risk monitoring [18, 54]. A reduction of professionals' workloads could be achieved if women participated more actively (e.g. self-collection of saliva samples and self-completion of data) under a nurses' supervision [43, 55]. In this line, our health professionals considered that their involvement would be facilitated if the current screening program were restructured, incorporating an adequate relationship with the health system and coordination with other levels of care.

## Health systems barriers and facilitators, for implementing personalized risk-based breast cancer screening

Implementing personalized screening will require substantial organizational changes to the current healthcare model [16, 50]. Despite the need for health interventions to evolve when new evidence appears, substantial efforts will be required to abandon a screening program in

use for more than 30 years and to introduce a new one that still presents uncertainties. Health professionals are used to changes in clinical protocols as new studies support them, but they are reluctant to face organizational changes whose dimension and effects they do not control. Rainey et al. suggest that countries that already have an organized, population-based screening program in place may experience fewer difficulties implementing a risk-based screening program [16], and professionals in our study consider that the experience with other risk-based interventions would reduce reluctance to change. Likewise, Bellhouse et al. [41] conclude that the integration of risk assessment and management tools into practice software or access to web-based applications would facilitate this change, as happened for cardiovascular risk assessment and management.

Professionals participating in our study indicated that risk-based screening would only be possible if the current IT tools were updated. Similarly, Puzhko et al. [14] emphasize the importance of integrating the risk calculator into the electronic medical record and Yi et al. [18] emphasize the need for developing electronic decision aids to enhance risk communication. Likewise, information systems must integrate electronic medical record and screening information in an easy and accessible way for all stakeholders. This would facilitate monitoring the risk of breast cancer overtime, accounting for potential changes in risk factors [16, 18].

Decision aids are needed to allow participatory health models such as shared decision-making [14]. Decision aids allow communication of the risks and benefits of screening in a balanced way, and work together to reach a decision about adherence to the program, based on clinical evidence and the patient's informed preferences [16, 54]. Studies show that women participating in shared decision-making have a greater commitment to health decisions, since they feel more responsible for their actions. This would allow women to reduce barriers related to anxiety, decisional conflict, and to prefer more beneficial and less invasive options [28, 56, 57], that is, to better manage the expectations of a risk-based program [16]. We included shared decision-making in our discussion group guide as an innovative strategy in screening. We were interested in knowing the views of health professionals about their implementation, considering that some barriers that have been reported in the literature have their origin in professionals' resistance to change [58].

Our study participants argued that the costs of updating the IT system and implementing shared decision-making, together with the aforementioned costs of genetic determinations and human resources for risk communication, will lead to budget increases. However, there is evidence in favor of risk-based screening being more efficient [3, 51–53], providing more resources to women at higher risk and fewer to those at lower risk. Harkness et al. [52] showed that compared to age-based screening, not offering screening to women in the lowest tertile of risk could improve effectiveness, cost-effectiveness, and decrease overdiagnosis, whilst resulting in a small excess of breast cancer deaths. However, Vilaprinyo et al. [3], in a modeling study, found that risk-based screening could reduce breast cancer mortality. Therefore, an economic evaluation of risk-based screening at the local level should be performed.

Given that both structural and organizational changes as well as an increase in economic resources will be required, the implementation of risk-based screening cannot be carried out without strong conviction from policy makers, health care providers, and regional and national authorities, to develop patient-centered medicine [49, 59, 60]. Therefore, declaration of intent is not enough, it is necessary to develop specific strategies for the application of risk-based screening [59, 60].

Finally, the evidence also supports the consensus reached in our discussion groups; primary care should be the gateway to the new program [14, 16, 41, 50]. However, among the professionals in our study, there was no agreement on which professionals should be in charge of risk communication and risk counseling, although nurses and family doctors were the most

cited. In an European study [59], Dutch and Swedish women agreed that below average and average risk results can be relayed in a letter, while above average risk feedback should be done through a telephone or face-to-face consultation with a family doctor or a specialized nurse, with expert knowledge in the field. British women at below average to average risk were also satisfied receiving information on their risk in a letter, but women at moderate or high risk recommended the development of special women's clinics operated by specialized nurses, radiographers, radiologists, and gynecologists, integrating breast and cervical cancer screening. British women also signaled a need to educate professionals on all aspects of risk-based screening and prevention in order to prevent the provision of conflicting information, and also a need for pathways and protocols to standardize interaction between primary and secondary care providers to avoid individual variation [59]. In line with this result, Culver et al. [61] and Bellhouse et al. [41] found that primary care physicians require more training in risk assessment and risk communication before facing complex scenarios in high-risk women. This evidence preserves some elements of the current screening program, such as the delivery of information by letter and telephone follow-up. However major changes are needed in several elements, such as information systems or risk communication protocols, for the implementation of a personalized risk-based screening.

## Strengths and limitations

To our knowledge, this is the first study in Spain that intended to obtain a description of barriers and facilitators as well as a global assessment of the difficulties in implementing a risk-based breast cancer screening program, all from the perspective and experiences of health professionals working in the NHS. The views of the different stakeholders involved in the screening, diagnosis and treatment of breast cancer have provided valuable information to pave the way for the implementation of personalized breast cancer screening.

Our study has some limitations. First, collecting data from discussion groups may have produced a social desirability bias in the discourse [62]. However, the triangulation of information through observations and data with the participation of different professionals, in addition to contributing to a better understanding of the phenomenon, relativized this possible bias by corroborating the convergences, divergences and complementarity of data obtained in the discussion groups [63]. In addition, reflexivity was maintained during the research process with the participation of the entire research team and the use of information from field notes [64]. Second, the four discussion groups had a heterogenous profile that limited the saturation of specific perspectives or views. However, the research team considered that for most of the topics discussed the information obtained was sufficient to answer the research questions. Third, not all our findings may be generalizable to other health systems. Nonetheless, many of them may be transferable to health systems with population-based screening programs. Fourth, participants' perspectives on perceived barriers and facilitators may have been influenced by the way the research team presented the most relevant changes in risk-based screening, compared to the current screening: risk measurement and communication and the corresponding screening recommendations.

## Conclusions

Despite the barriers and limitations identified, study participants were supportive of the implementation of risk-based screening. They highlighted the benefits of risk-based screening, especially for women at high risk of breast cancer and those under 50 years of age. They stressed that allocating more resources to these women is not only beneficial for them, but also translates into a greater efficiency of the risk-based program compared to the current one.

However, the implementation of risk-based screening would pose a great challenge for the SpNHS due to organizational and administrative transformations, and also due to the need for commitment and political will. Other challenges come from the need for coordination between the different levels of care; improvements to information systems; education of health professionals on personalized screening, risk communication and shared decision-making; and dissemination of information through public campaigns, guaranteeing that it is accessible to the entire population. As facilitators of the implementation, the participants highlighted the experience of PCP in other screening programs; the relationship of trust and the closeness of these professionals with women; the awareness of women about breast cancer and early detection; and the increasingly proactive role of women in their health care.

More studies are needed that expand knowledge and understanding about the views of health professionals, health system administrators and those responsible for formulating public health policies, on the viability of risk-based screening for breast cancer and its future implementation; and also the acceptance of this strategy by women. Future studies should assess the efficiency and feasibility of implementing risk-based breast cancer screening in clinical practice in the SpNHS.

## Supporting information

**S1 Table. Discussion groups guide.**
(DOCX)

## Acknowledgments

We thank the health professionals who generously and actively participated in the discussion groups. We also thank JP Glutting for reviewing and editing the manuscript.

**The DECIDO Group**

Members of the DECIDO Study Group: Pau Balaguer-Llaquet; Iván-David Benítez; Alexandra Bertran; Àngels Cardona; Cristina Cazorla-Sánchez; Núria Codern; Inés Cruz-Esteve; Carles Forné-Izquierdo; Marta Hernández-Andreu; María José Hernández-Leal; Edelmir Iglesias; Gisela Galindo-Ortego; Celmira Laza-Vásquez; Montserrat Llorens-Gabandé; Montserrat Martínez-Alonso; Hèctor Perpiñán; Anna Pons-Rodríguez; Mercè Reñé-Reñé; Montserrat Rué; Isabel Sánchez-López; Jordi Vilaplana-Mayoral.

## Author Contributions

**Conceptualization:** Misericòrdia Carles-Lavila, Montserrat Rué.

**Data curation:** Núria Codern-Bové, Àngels Cardona-Cardona, Misericòrdia Carles-Lavila, Montserrat Rué.

**Formal analysis:** Celmira Laza-Vásquez, Núria Codern-Bové, Àngels Cardona-Cardona.

**Investigation:** Misericòrdia Carles-Lavila, Montserrat Rué.

**Methodology:** Núria Codern-Bové, Àngels Cardona-Cardona, Misericòrdia Carles-Lavila, Montserrat Rué.

**Software:** Núria Codern-Bové, Àngels Cardona-Cardona.

**Validation:** Celmira Laza-Vásquez, Núria Codern-Bové, Àngels Cardona-Cardona, Maria José Hernández-Leal, Maria José Pérez-Lacasta, Misericòrdia Carles-Lavila, Montserrat Rué.

**Visualization:** Celmira Laza-Vásquez, Núria Codern-Bové, Àngels Cardona-Cardona, Maria José Hernández-Leal, Maria José Pérez-Lacasta, Misericòrdia Carles-Lavila, Montserrat Rué.

**Writing – original draft:** Celmira Laza-Vásquez, Núria Codern-Bové, Àngels Cardona-Cardona, Maria José Hernández-Leal, Maria José Pérez-Lacasta, Misericòrdia Carles-Lavila, Montserrat Rué.

**Writing – review & editing:** Celmira Laza-Vásquez, Núria Codern-Bové, Àngels Cardona-Cardona, Maria José Hernández-Leal, Maria José Pérez-Lacasta, Misericòrdia Carles-Lavila, Montserrat Rué.

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
