## [Decision Letter · Decision Letter 0]

16 Nov 2021

PONE-D-21-21753Views of health professionals on risk-based breast cancer screening and its implementation in the Spanish National Health System: A qualitative discussion group studyPLOS ONE

Dear Dr. Rue,

Thank you for submitting your manuscript to PLOS ONE. After careful consideration, we feel that it has merit but does not fully meet PLOS ONE’s publication criteria as it currently stands. Therefore, we invite you to submit a revised version of the manuscript that addresses the points raised during the review process.

We look forward to receiving your revised manuscript.

Kind regards,

Eugenio Paci, MD

Academic Editor

PLOS ONE

Journal Requirements:

2. When reporting the results of qualitative research, we suggest consulting the COREQ guidelines: http://intqhc.oxfordjournals.org/content/19/6/349. In this case, please consider including more information on the number of interviewers, their training and characteristics. Moreover, please provide the interview guide used as a Supplementary File.

“This study was supported by the research grants ‘Personalized breast cancer screening: assessment of its feasibility and acceptability in the National Health System’ (PI17/00834) and ‘Collaboration of healthcare professionals to include shared decision-making in the breast cancer screening program’ (PI18/00773) from the Instituto de Salud Carlos III and cofunded by Fondo Europeo de Desarrollo Regional (FEDER) ‘Una manera de hacer Europa’. Celmira Laza Vásquez received a grant from Santander Program scholarship 2020 as a predoctoral fellow at the University of Lleida. The funders did not participate in the study design; collection, management, analysis, and interpretation of data; writing of the report; and the decision to submit the report for publication.”

“MR

PI17/00834: Personalized breast cancer screening: assessment of its feasibility and acceptability in the National Health System. Instituto de Salud Carlos III and cofunded by Fondo Europeo de Desarrollo Regional (FEDER) ‘Una manera de hacer Europa’https://www.isciii.es

MCL

PI18/00773: Collaboration of healthcare professionals to include shared decision-making in the breast cancer screening program.Instituto de Salud Carlos III and cofunded by Fondo Europeo de Desarrollo Regional (FEDER) ‘Una manera de hacer Europa’ https://www.isciii.es

CLV Santander Program scholarship 2020:  predoctoral fellow at the University of Lleida

The funders did not participate in the study design, data collection and analysis, decision to publish, or preparation of the manuscript.”

Additional Editor Comments (if provided):

This is an important qualitative research considering the main aspects and problems related to the future, possible implementation of personalized approaches in cancer screening. The presentation is interesting with reference to the debate on this issue. Perhaps the manuscript is long. I will suggest to the authors to reduce the length, if possible. The discussion session considered as one of the most relevant issue the reduction of the interval in lower risk women. This attitude is certainly a manifestation of fear and uncertainty. However, I invite to consider the case of cervical cancer screening where the change from pap-smear to Hpv-testing gave the possibility to change from 3 to 5 years interval. The change of type of technology as a new, innovative screening test possibly facilitated the acceptance of a longer interval.

Reviewers' comments:

Reviewer's Responses to Questions

**Comments to the Author**

1. Is the manuscript technically sound, and do the data support the conclusions?

Reviewer #1: Yes

Reviewer #2: Yes

2. Has the statistical analysis been performed appropriately and rigorously? 

Reviewer #1: N/A

Reviewer #2: N/A

3. Have the authors made all data underlying the findings in their manuscript fully available?

Reviewer #1: Yes

Reviewer #2: Yes

4. Is the manuscript presented in an intelligible fashion and written in standard English?

Reviewer #1: No

Reviewer #2: Yes

5. Review Comments to the Author

Reviewer #1: General comments

The paper affords an interesting and important issue with the appropriate method.

In general, the paper can be shortened since many parts of the introduction are reported similarly in the discussion; also the results are redundant in some parts of the text that simply repeats the quotes reported in the tables.

I suggest to check better the classification of quotes in the four tables and also their relevance for the “barriers” and “facilitators”: in some cases I found the classification arbitrary or even incorrect.

Please check the text: table 2 is placed in the wrong place and table 1S is missing.

There are many typos and incomplete sentences; I cannot be sure since I am not native English, but probably there are also some grammar errors (for example nouns used as adjectives should be singular, the use of comas and/or “and” in list is misleading).

Specific comments

Abstract

Results: the list of facilitators is a mix of existing factors (that should be used or emphasized) and some actions or interventions that should be present but are not in place now (example commitment of policy makers; training of health care workers). Can you try to write the sentence to explain it?

Conclusion: the sentence “to our knowledge…” (lines 51-3) is not useful for an abstract (I suggest to drop it even in the text).

Introduction

It can be shortened, for example, the paragraph from line 112 to line 135 can be dropped since it is reported in the discussion in the same way; also lines 138-141 are in the discussion. Why are you introducing shared decision making? I did not find this topic in the discussion groups (maybe because table S1 is missing).

Line 89: “screening personalization is effective” I suggest to change the sentence in “models forecast that personalised screening may increase health benefits and be more efficient”

Line 150: the intervention you are proposing to the discussion would be not “clinical practice”, but preventive practice.

Methods

Line 193-195: please explain why a focus group does not allow positioning of different profiles.

Line 197: I suppose “table 2” should be table S1 (that is missing in the pdf), while table two refers to the first paragraph of the results.

Lines 239-242: is this part necessary? This sentence repeats the objective.

Results

In the first paragraph, please introduce also the final subheading about the “organizational proposals”, this would help the reader.

Table 2: first and second barriers seem the same to me. Furthermore, this issue may be also a problem related to the health professionals. “facilitators: benefits of personalized screening” the point of younger women is much more specific and deserves a specific title.

table 4: “cost of measuring the risk” In the quotes I see costs of measuring and communicating risk and of surveillance.

table 5: “budget allocation” why is this different from the costs and sustainability of the program? I do not see a cost opportunity point of view in these quotes, but they are rather similar to that in table 4 about costs

please check again the classification of the sentence in your research group

Line 318: I suggest to name this paragraph using the same words used in the initial paragraph of the results where you introduced it.

Discussion

lines 536-537: the predictive value of available SNPs is different in different ancestries, see Liu C eta alt JAMA network open 2021 and Du Z et al JNCI 2021.

Conclusion

Lines 667-670: I suggest to drop it at all, but definitely it is not a conclusion.

Reviewer #2: This study explores the barriers and facilitators, as perceived by healthcare professionals, for implementing risk-based breast screening programme in Spain.

This is a timely topic. Overall the study is well thought and clearly written. I enjoyed reading it.

Few minor points:

To understand the context and have an insight into the responses of the participants, it will be very helpful to add:

• short introduction to the organization of the existing age-based screening programme, explaining the role of the primary care physicians

• the information the participants were provided about risk-based screening

For example, on p19 and p31, ‘…since the potential implementation of risk-based breast screening will require extensive changes in current practice…’ and ‘…substantial efforts will be required to abandon a screening programme in use for more than 30 years and introduce a new one that still presents uncertainties…’ So what is the current structure and what changes are anticipated with risk-based screening? If the expected changes are assumed changes, and presented to the participants as changes needed, then this would influence the perspective of the participants on the perceived barriers and facilitators.

Looking into some of responses quoted such as ‘…genomic test cannot be performed at home by the user, so they need assistance’ [DG2P3]; ‘…in large European populations they (SNPS) are not yet validated’ [DG3P6]; ‘…all women over 40 years have to do a mamogram’[DG1P3], etc, it is not clear whether these reflect the information provided to the participants or misconceptions of the participants (e.g. genetic testing). If the latter, then in itself informative that not knowing much about the field generates fear and resistance to change. Worth the authors expanding on these points.

The category of women’s perceptions is interesting, it reflects the perception of the HCPs of the perception of women. In fact, how much these reflect HCPs own perceptions and their concerns but articulated from women’s point of view.

Helpful to expand if there was any discussion of the HCPs perception of ‘how to’ overcome the barriers, how to get policy commitment?

It’s unfortunate that the question on the perception of the HCPs on how should the personalised screening programme to be organized.

P29, ‘…there is consensus on incorporating personal and family history, breast density, benign breast disease and lifestyle factors…’ Is there a consensus to incorporate all these risk factors in population-based screening programme or is there evidence that comprehensive risk assessment model improves the risk prediction?

In the Discussion section, sometimes it is not clear what the authors are recommending based on the findings vs. based on their summary of the literature. As one of the several examples, ‘Thus, health professionals seemed to have a good predisposition towards risk-based screening. However, it is essential that they know the results of the research and understand the statistics of early detection to avoid misleading and potentially harmful recommendations [18].’

6. PLOS authors have the option to publish the peer review history of their article (what does this mean?). If published, this will include your full peer review and any attached files.

Reviewer #1: **Yes: **paolo giorgi rossi

Reviewer #2: No

---

## [Author Response · Author response to Decision Letter 0]

24 Jan 2022

The authors would like to thank the editor and reviewers for their comments, as their contributions have considerably improved the new version. Suggested modifications have been made to the text, which are highlighted in the file 'Revised Manuscript with Track Changes'. You can find a specific answer to each comment below:

Editorial team We note that you have provided additional information within the Acknowledgements Section that is not currently declared in your Funding Statement. Please note that funding information should not appear in the Acknowledgments section or other areas of your manuscript. We will only publish funding information present in the Funding Statement section of the online submission form.

“MR: PI17/00834: Personalized breast cancer screening: assessment of its feasibility and acceptability in the National Health System. Instituto de Salud Carlos III and cofunded by Fondo Europeo de Desarrollo Regional (FEDER) ‘Una manera de hacer Europa’ https://www.isciii.es

MCL: PI18/00773: Collaboration of healthcare professionals to include shared decision-making in the breast cancer screening program.Instituto de Salud Carlos III and cofunded by Fondo Europeo de Desarrollo Regional (FEDER) ‘Una manera de hacer Europa’ https://www.isciii.es

CLV: Santander Program scholarship 2020: predoctoral fellow at the University of Lleida

MJH-L: European Regional Development Fund (ERDF). European Union’s Horizon 2020 research and innovation programme under the Marie Skłodowska-Curie grant agreement No. 713679 from the Universitat Rovira i Virgili (URV).

The funders did not participate in the study design, data collection and analysis, decision to publish, or preparation of the manuscript.”

Answer: We have added a source of funding for Maria José Hernández-Leal. We forgot to include it in the first submission. Otherwise, we agree on the above specified Funding Statement and we have removed the study’s funding information from the manuscript.

Editorial team 4. Please review your reference list to ensure that it is complete and correct. If you have cited papers that have been retracted, please include the rationale for doing so in the manuscript text, or remove these references and replace them with relevant current references. Any changes to the reference list should be mentioned in the rebuttal letter that accompanies your revised manuscript. If you need to cite a retracted article, indicate the article’s retracted status in the References list and also include a citation and full reference for the retraction notice.

Answer: The entire reference list has been reviewed and no retracted papers have been cited.

Additional Editor Comments:

This is an important qualitative research considering the main aspects and problems related to the future, possible implementation of personalized approaches in cancer screening. The presentation is interesting with reference to the debate on this issue. Perhaps the manuscript is long. I will suggest to the authors to reduce the length, if possible. The discussion session considered as one of the most relevant issue the reduction of the interval in lower risk women. This attitude is certainly a manifestation of fear and uncertainty. However, I invite to consider the case of cervical cancer screening where the change from pap-smear to Hpv-testing gave the possibility to change from 3 to 5 years interval. The change of type of technology as a new, innovative screening test possibly facilitated the acceptance of a longer interval.

Answer: We have reduced the length of the manuscript. In the introduction, we have supressed several lines in the paragraph that starts with “Several studies have addressed the perspectives…”. Nevertheless, we have added some sentences to clarify some aspects that the reviewers rose up (e.g. the organization of screening in Spain or the role of primary care physicians).

With respect ot the case of cervical cancer, we have added the following sentences at the end of the third paragraph in the Barriers and facilitators related to women subsection of the Discussion: A research synthesis on factors associated to the acceptability of human papillomavirus HPV testing for cervical cancer found that strategies that increase women’s knowledge and efforts to increase health care providers’ awareness might also increase acceptability of wider screening intervals (Tatar, 2018).

Reviewers’ comments 

Reviewer #1 (R#1): General comments The paper affords an interesting and important issue with the appropriate method. In general, the paper can be shortened since many parts of the introduction are reported similarly in the discussion; also the results are redundant in some parts of the text that simply repeats the quotes reported in the tables.

Answer: Thank you, very much. The introduction has been shortened. We have eliminated some of the quotes in the tables because of redundancy.

R#1 I suggest to check better the classification of quotes in the four tables and also their relevance for the “barriers” and “facilitators”: in some cases I found the classification arbitrary or even incorrect.

Answer: The reviewer is right. We have reviewed the tables and made changes in order to increase the correspondence between them and the assigned barriers and facilitators. In some cases it is difficult, because the same quote refers to more than one theme.

R#1 Please check the text: table 2 is placed in the wrong place and table 1S is missing.

Answer: Thank you, we have moved Table 2 to the right place and added table S1.

R#1 There are many typos and incomplete sentences; I cannot be sure since I am not native English, but probably there are also some grammar errors (for example nouns used as adjectives should be singular, the use of comas and/or “and” in list is misleading).

Answer: The new version of the manuscript has been revised by a native English speaker.

R#1 Specific comments Abstract Results: the list of facilitators is a mix of existing factors (that should be used or emphasized) and some actions or interventions that should be present but are not in place now (example commitment of policy makers; training of health care workers). Can you try to write the sentence to explain it?

Answer: The last sentence in the results section in the abstract now states that the listed interventions will be required. We have moved “commitment of policy makers” from the previous sentence, which includes actions or interventions that are in place now or that there is evidence, to the last sentence.

R#1 Conclusion: the sentence “to our knowledge…” (lines 51-3) is not useful for an abstract (I suggest to drop it even in the text).

Answer: The sentence has been eliminated from the abstract and moved to another paragraph in the Discussion section, in the text.

R#1 Introduction It can be shortened, for example, the paragraph from line 112 to line 135 can be dropped since it is reported in the discussion in the same way; also lines 138-141 are in the discussion. Why are you introducing shared decision making? I did not find this topic in the discussion groups (maybe because table S1 is missing).

Answer: The introduction has been shortened. The paragraph from line 112 to line 135 has been summarized. We introduced shared decision-making because we think that personalized screening needs the consensus of women and health professionals. As the reviewer suggests the fact that table S1 was missing, made it difficult to understand why we introduced shared decision-making.

R#1 Line 89: “screening personalization is effective” I suggest to change the sentence in “models forecast that personalised screening may increase health benefits and be more efficient”

Answer: We agree with the reviewer. We have modified the sentence: “… with personalized screening, the gain in quality adjusted life years would be higher at a lower cost, compared with the standard one-size-fits-all strategy.”

R#1 Line 150: the intervention you are proposing to the discussion would be not “clinical practice”, but preventive practice.

Answer: We use “clinical practice” in the broad sense. There are many clinical guidelines on screening, as well as sections in journals called clinical practice that include articles on screening.

R#1 Methods Line 193-195: please explain why a focus group does not allow positioning of different profiles.

Answer: In the second paragraph of the Data Collection section we have explained the advantages of discussion groups when exploring the views of health professionals on risk based breast cancer screening and its implementation in the Spanish NHS. We have intended to present the discussion group technique as an open and less directive form of the inquiry than the focus group.

R#1: Line 197: I suppose “table 2” should be table S1 (that is missing in the pdf), while table two refers to the first paragraph of the results.

Answer: The reviewer is correct, we forgot to submit Table S1 and we put Table 2 in the wrong place. We have corrected both errors in the new version of the manuscript.

R#1: Lines 239-242: is this part necessary? This sentence repeats the objective.

Answer: We have deleted the paragraph.

R#1 Results In the first paragraph, please introduce also the final subheading about the “organizational proposals”, this would help the reader.

Answer: Done

R#1 Table 2: first and second barriers seem the same to me.

Answer: The reviewer is right. Both barriers are similar, although rejecting recommendations does not imply necessarily resistance to change. We have changed the quotes in the barrier “resistance to change”.

R#1 Furthermore, this issue may be also a problem related to the health professionals. “facilitators: benefits of personalized screening” the point of younger women is much more specific and deserves a specific title.

Answer: We have added “for young women” in the title "benefits of a ris-based program, in table 2.

R#1 table 4: “cost of measuring the risk” In the quotes I see costs of measuring and communicating risk and of surveillance.

Answer: We have shortened the second quote and maintained the title of the section.

R#1 table 5: “budget allocation” why is this different from the costs and sustainability of the program? I do not see a cost opportunity point of view in these quotes, but they are rather similar to that in table 4 about costs please check again the classification of the sentence in your research group

Answer: The reviewer is right. We have changed the quotes so now they seem more adequate to us.

R#1 Line 318: I suggest to name this paragraph using the same words used in the initial paragraph of the results where you introduced it.

Answer: We have changed the section title to “Barriers and facilitators related to implementing a risk-based screening program”.

R#1 Discussion lines 536-537: the predictive value of available SNPs is different in different ancestries, see Liu C eta alt JAMA network open 2021 and Du Z et al JNCI 2021.

Answer: We have added a sentence that includes both references.

R#1 Conclusion Lines 667-670: I suggest to drop it at all, but definitely it is not a conclusion.

Answer: We have moved it to the Strengths and Limitations section of the Discussion.

Reviewer #2 (R#2) This study explores the barriers and facilitators, as perceived by healthcare professionals, for implementing risk-based breast screening programme in Spain.

This is a timely topic. Overall the study is well thought and clearly written. I enjoyed reading it.

Answer: Thank you very much.

R#2 Few minor points: To understand the context and have an insight into the responses of the participants, it will be very helpful to add:

• short introduction to the organization of the existing age-based screening programme, explaining the role of the primary care physicians 

• the information the participants were provided about risk-based screening

For example, on p19 and p31, ‘…since the potential implementation of risk-based breast screening will require extensive changes in current practice…’ and ‘…substantial efforts will be required to abandon a screening programme in use for more than 30 years and introduce a new one that still presents uncertainties…’ So what is the current structure and what changes are anticipated with risk-based screening? If the expected changes are assumed changes, and presented to the participants as changes needed, then this would influence the perspective of the participants on the perceived barriers and facilitators.

Answer: 1) A brief introduction to the existing screening program in Spain has been added to the 6th paragraph of the introduction. We also have included a new paragraph in the introduction (8th) that describes the recommendations of the Spanish Family and Community Medicine for primary care physicians.

2) Participants received information about risk-based screening before starting the discussion. The supplementary information file, that we forgot to append when we submitted the manuscript, contains the details of the discussion guide. There, we say that two members of the research group presented a proof of concept study on the feasibility and acceptability of risk-based screening, carried out by the study team. We have added a sentence with the details in section Data Collection of the manuscript with a reference to the study protocol.

The reviewer is right when she or he says that the perspective of the participants on perceived barriers and facilitators may have been influenced by how changes needed for risk-based screening were presented. We have added a limitation based on this comment in the Discussion section.

R#2 Looking into some of responses quoted such as ‘…genomic test cannot be performed at home by the user, so they need assistance’ [DG2P3]; ‘…in large European populations they (SNPS) are not yet validated’ [DG3P6]; ‘…all women over 40 years have to do a mamogram’[DG1P3], etc, it is not clear whether these reflect the information provided to the participants or misconceptions of the participants (e.g. genetic testing). If the latter, then in itself informative that not knowing much about the field generates fear and resistance to change. Worth the authors expanding on these points.

Answer: We think that both the information provided -which was scarce due to time limitations- and some misconceptions of the participants contributed to generate fear and resistance to change. A recent systematic review by Bellhouse et al. indicates that primary care providers when assessing breast cancer risk focused on collection of family history and provision of support for women at increased risk. The authors mention that insufficient education/training and perceived discomfort were the most commonly endorsed barriers reported for risk assessment and risk-reducing medication. Nevertheless, the integration of risk assessment and management tools into practice software or access to web-based applications would facilitate the change, as happened for cardiovascular risk assessment and management.

We have included these findings in different paragraphs of the discussion.

R#2 The category of women’s perceptions is interesting, it reflects the perception of the HCPs of the perception of women. In fact, how much these reflect HCPs own perceptions and their concerns but articulated from women’s point of view.

Answer: Reviewer #2 is right. We have not measured Spanish women’s views on risk-based screening and therefore we don’t know how much HCP’s perceptions reflect those of women. But, when Rainey et al. (2018) explored whether professionals views of acceptability of risk-based breast cancer screening and prevention were in line with those of eligible women, in the United Kingdom, Sweden and the Netherlands, the answer was positive. Also, European women’s perceptions of the implementation and organisation of risk-based breast cancer screening and prevention, were consistent with those of professionals (Rainey 2020).

We have slightly modified a couple of sentences in this section of the Discussion, to highlight this agreement.

R#2

Helpful to expand if there was any discussion of the HCPs perception of ‘how to’ overcome the barriers, how to get policy commitment?

Answer: Participants, during the discussion, proposed several facilitators to overcome the barriers. Unfortunately, some of the issues, such as how to achieve political compromise, were not discussed in sufficient depth due to lack of time.

R#2

It’s unfortunate that the question on the perception of the HCPs on how should the personalised screening programme to be organized.

Answer: Yes, that is true.

R#2 P29, ‘…there is consensus on incorporating personal and family history, breast density, benign breast disease and lifestyle factors…’ Is there a consensus to incorporate all these risk factors in population-based screening programme or is there evidence that comprehensive risk assessment model improves the risk prediction?

Answer: The reviewer made a good point. Probably there is not yet consensus on incorporating all these risk factors in population-based screening programs. Since a few lines before we had written "The network highlights how risk stratification has improved with the use of comprehensive models ...", we have shortened the sentence to: "Although research is still ongoing to assess the clinical utility of PRS for population screening programs, Yanes et al. point out that polygenic testing is already being implemented in specialist familial cancer clinics...".

R#2 In the Discussion section, sometimes it is not clear what the authors are recommending based on the findings vs. based on their summary of the literature. As one of the several examples, ‘Thus, health professionals seemed to have a good predisposition towards risk-based screening. However, it is essential that they know the results of the research and understand the statistics of early detection to avoid misleading and potentially harmful recommendations [18].’

Answer: We have tried to clarify which recommendations come from our study and which come from the literature.

---

## [Editor Report · Decision Letter 1]

27 Jan 2022

Views of health professionals on risk-based breast cancer screening and its implementation in the Spanish National Health System: A qualitative discussion group study

PONE-D-21-21753R1

Dear Dr. Rue,

We’re pleased to inform you that your manuscript has been judged scientifically suitable for publication and will be formally accepted for publication once it meets all outstanding technical requirements.

Kind regards,

Eugenio Paci, MD

Academic Editor

PLOS ONE

Additional Editor Comments (optional):

Thank you for the work done. All comments have been addressed.
---

## [Editor Report · Acceptance letter]

28 Jan 2022

PONE-D-21-21753R1 

Views of health professionals on risk-based breast cancer screening and its implementation in the Spanish National Health System: A qualitative discussion group study 

Dear Dr. Rué:

I'm pleased to inform you that your manuscript has been deemed suitable for publication in PLOS ONE. Congratulations! Your manuscript is now with our production department. 

Kind regards, 

on behalf of

Dr. Eugenio Paci 

Academic Editor

PLOS ONE